# Adverse events following COVID-19 vaccination in Kwara State, North-central Nigeria

Louis Okeibunor Odeigah[1], Yahkub Babatunde Mutalub[2,3]*, Olalekan Ayodele Agede[4], Ismail A. Obalowu[5], Susan Aiyetoro[6], Gafar A. A. Jimoh[7]

1 Department of Family Medicine, College of Health Science University of Ilorin, Ilorin, Kwara State, Nigeria, 2 Department of Clinical Pharmacology and Therapeutics, College of Medical Sciences, Abubakar Tafawa Balewa University, Bauchi, Nigeria, 3 Department of Family Medicine Abubakar Tafawa Balewa University Teaching Hospital, Bauchi, Nigeria, 4 Department of Pharmacology and Therapeutics University of Ilorin, Ilorin, Nigeria, 5 Department of Family Medicine, General Hospital Ilorin, Ilorin, Kwara State, Nigeria, 6 Department of Pharmacy and Zonal Pharmacovigilance Center, University of Ilorin Teaching Hospital, Ilorin, Nigeria, 7 Department of Obstetrics and Gynaecology, University of Ilorin, Ilorin, Nigeria

* bymutalub@atbu.edu.ng, yahkubmutalub@gmail.com

**Data Availability Statement:** All data are included in the paper and the Supporting Information file titled S1 Dataset.

## Abstract

Safe and effective vaccination remains the mainstay of control of COVID-19 because there is still no universally recommended treatment. This strategy is however being threatened by vaccine hesitancy and resistance due to fear of adverse events and safety concerns. It is, therefore, necessary to study post-vaccination adverse events (AE) in various populations and geographical areas. The objective of this study was to analyze the adverse events following COVID-19 vaccination in five major immunization centers of Kwara State Nigeria. A retrospective descriptive study of the adverse events following AstraZeneca COVID-19 vaccination that were reported from five immunization centers of Kwara State, North-central Nigeria from March to July 2021 was carried out. Statistical Package for Social Science version 26 was used for analysis. Adverse event classification and severity were compared based on age, gender, and time to onset of adverse event and vaccine dose type using the Chi-square test. The incidence of COVID-19 vaccine AE was 1.6%. There was female predominance (51.6%) and a mean age of 41.6±13.7 years. Most of the AE (95.8%) were systemic and mild (81.1%) without a requirement for any therapeutic intervention. Fatal outcome was not reported in any of the AE and the time to outcome of AE was 2 days in most cases (45.3%). No significant association was found between the variables studied and the adverse event type and severity. The low incidence and mild nature of adverse events reported in this study will add to the body of knowledge regarding vaccine adverse events and may eventually impact vaccine uptake.

## Introduction

The Coronavirus Disease 2019 (COVID-19) was declared a global pandemic by the World Health Organisation (WHO), on the 11th of March 2020 [1]. It has been associated with

**Funding:** The authors received no specific funding for this work.

**Competing interests:** The authors have declared that no competing interests exist.

significant morbidity and mortality as well as negative effects on the economy and human socialization which have impacted the quality of life and psycho-social health of individuals [2]. According to the WHO, there were 376,478,335 confirmed cases and 5,666,064 individuals have died globally from COVID-19 as of 1st February 2022 [3]. In Nigeria, a total of 253,340 confirmed cases and 3,136 deaths from COVID-19 was reported as of 1st February 2022 [4].

Researchers globally have been battling with the challenges of combating the epidemic for over 2 years through re-directing existing drugs and new drugs as well as vaccine discovery [5]. Even though several therapeutic agents have been tried for the treatment of COVID-19, there is still no universally recommended one. Efforts are therefore geared toward its prevention to control the spread and reduce the burden on the healthcare system [6]. Hence, the formulation of effective and safe vaccines remains the mainstay to ameliorate this pandemic [7].

Although one of the most effective ways of controlling communicable diseases like COVID-19 is through vaccination, successful vaccination is often challenged by hesitancy and resistance due to fear of adverse events and safety concerns of the vaccine [8]. The WHO (2019) identified vaccine hesitancy as one of the top 10 global health threats [9]. The identified cause of vaccine hesitancy is major doubts about the safety and efficacy of the vaccines and misinformation about the COVID vaccine [8]. Skepticism about COVID-19 vaccines due to lack of trust in vaccine safety is high in West African countries according to Afrobarometer survey [8].

Fear of vaccine adverse events and safety concerns are reasons for vaccine refusal cited in various studies [6,10]. A massive amount of information regarding the COVID-19 vaccine including several of those that are not evidence-based has rapidly spread worldwide. This has led to the dissemination of COVID-19 myths and conspiracies [11]. For COVID-19 vaccination to be globally successful, the conspiracy theories have to be demystified totally, and the issue of the serious COVID-19 vaccine adverse events should be resolved while more studies to support what is likely and what is unlikely are conducted [12].

About thirteen COVID-19 vaccines including AstraZeneca viral vector vaccine are granted emergency approval in various countries [5]. The WHO has recommended the AstraZeneca COVID-19 vaccine for use in all adults. It has been recommended two doses of the vaccine be given 8 to12 weeks apart [13]. It has proven efficacy and safety as revealed by the outcome of clinical trials in South Africa, Brazil, and the United Kingdom [14]. AstraZeneca revealed that the vaccine was 100% effective in preventing severe disease and hospital admission according to advanced trial data. The vaccine was also said to be 79% effective with little or no risk of formation of blood clots [15]. As of 30th January 2022, a total of 9,901,135,980 vaccine doses have been administered globally [3]. Nigeria received 3,924,000 Oxford/AstraZeneca COVID-19 vaccine doses through COVID-19 Vaccines Global Access Facility (COVAX) on March 2, 2021. Covid-19 vaccination in Nigeria therefore commenced on March 5, 2021 [16].

The routine procedure for new drug and vaccine approval were bypassed in giving emergency approval for COVID vaccines as a result of compelling necessity to prevent the continued spread of COVID-19 infection. Hence the need for post-vaccination adverse event monitoring [5]. Nevertheless, the safety of this vaccine has been well-studied, the pattern and extent of adverse events from vaccinations may differ based on ethnicity and region, Also the likelihood of undocumented adverse events is a consideration [14,17,18]. It is, therefore, necessary to study post-vaccination adverse events in various populations and geographical areas.

Despite the serious commitment of governments and health institutions to quickly achieve herd immunity by COVID-19 vaccination, information especially from social media on adverse events following intake of the vaccine has resulted in a considerable level of fear among the general population in Nigeria hence, jeopardizing the achievement of the vaccination program [16]. Several adverse events associated with COVID-19 vaccines have been

reported especially among healthcare workers [12,19–21]. However, there has been a paucity of data on adverse events following immunization with the COVID-19 vaccine among the general population in our community. As a result, we examined all of the AstraZeneca COVID-19-vaccine-related adverse events recorded in Kwara State's five major immunization centers during the study period. This will therefore provide a safety signal for COVID-19 vaccinations at an early stage and serve as a foundation for future research into the vaccine's safety data.

## Methods

### Study design and population

This was a retrospective descriptive study of the adverse events following AstraZeneca COVID-19 vaccination reported from five immunization centers of Kwara State, North-central Nigeria from March to July 2021. AstraZeneca COVID-19 vaccine was the only available vaccine in the state; hence, it was the only one administered during the study period. The study participants were all individuals who received the Covid-19 vaccine in the five studied vaccination centers in Kwara state between March and July 2021. The first COVID-19 vaccination exercise in Kwara state was conducted from 17[th] to 26[th] March 2021, the second one was from 3[rd] to 12[th] May 2021 and the third exercise was from 22[nd] June to 3[rd] July 2021. The report of adverse events from primary health care center Idofian, civil service Hospital, University of Ilorin immunization center as well as University of Ilorin Teaching Hospital immunization centers in Oke-oyi and Amilegbe was analyzed. All individuals receiving the vaccine were asked to report any adverse reactions post-vaccination to the immunization center or the emergency room if the adverse events were serious. Those who reported adverse events were monitored by twice-daily phone calls and home visits and were followed up for 3months after vaccination. Prophylactic use of antipyretics was not recommended but allowed depending on personal health conditions.

### Data collection and outcomes

Information was collected on the individual's, gender, age, vaccine dose, date of vaccination, date of presentation due to the adverse event, use of antipyretic drugs, other interventions, and outcome of the adverse effects. Information collected on specific adverse events included local and systemic reactions. Local adverse events included injection-site pain, itching, redness, swelling, and induration. Systemic adverse events included fever, headache, generalized body weakness/pain, malaise, muscle ache, joint pain, fatigue, dizziness, chills/rigor, vomiting, diarrhea, abdominal pain, and changed mental status. The severity of adverse events was graded as mild, moderate, and severe according to WHO criteria, as modified by Bae et al [17]. Outcomes of adverse events were classified into full resolution, resolution with a sequel, and with fatality.

### Statistical analysis

Descriptive statistics were reported in the form of frequency and percentage. Statistical Package for Social Science version 26 was used for analysis. Adverse event classification and severity were compared based on age, gender, and time to onset of adverse event and vaccine dose type using the Chi-square test. P values less than 0.05 were considered statistically significant.

### Ethical approval

This study was reviewed and approved by the Ethical Review Committee of the University of Ilorin Teaching Hospital with Approval Number: ERCPAN/2021/10/0204. Informed consent was waived because of the retrospective nature of the study.

## Results

A total of 5816 individuals were vaccinated out of whom 95 reported adverse events (AE) during the study period. Therefore, the incidence of COVID-19 adverse events in the study population was 1.6%. The reported AE were from individuals with a predominance of females (51.6%) and a mean age of 41.6±13.7 years with most belonging to the age group of 30–39 years (24.2%). The majority of the AE reported were among the recipients of the first vaccine dose (90.5%). Although most of the AE (58.9%) had its onset within the first day of vaccination, 5.3% had it within 6 hours of vaccination (Table 1).

In terms of AE type, almost all of the reported events (95.8%) occurred as systemic with or without local reaction, 35.8% as both local and systemic reactions while local with or without systemic reaction was reported in 40% of cases whereas, 4.4% of the reported AE occurred as a local reaction only (Table 2). Considering the severity, above four-fifths of the report (81.1%) were mild with only 2.1% being severe. Of the two severe AE, one was serious and life-threatening necessitating resuscitation and subsequent hospital admission. Moreover, in terms of intervention, the majority of the AE (80%) were observed without a requirement for any therapeutic intervention while 2.1% required hospital admission. Fatal outcome was not reported in any of the AE as almost all of them (98.9%) had full recovery without sequelae. The time to the outcome of AE was 2 days in most of the cases (45.3%) while 2.1% of the cases had their outcome in one week and above (Table 2).

In the local reaction category, injection-site pain occurred in all the individuals while itching and rashes at the injection site occurred in 2.6% of those with local reactions. The most commonly reported systemic reactions were headache (49, 51.6%), fever (45, 49.5%), generalized body weakness (37, 40.7%), and generalized body pain (21, 23.1%) while a noticeably low proportion of individuals reported muscle pain (1.1%), chest pain (2.2%) and dizziness (3.3%) (Table 3).

Table 4 shows no significant association between the age, gender, vaccine dose as well as time to onset of AE and the reported adverse event categories in terms of local and systemic reactions (p-value greater than 0.05 in all cases). Also, no significant association was found between the adverse event severity and age, gender, vaccine dose as well as time to onset of adverse effect (p-value greater than 0.05 in all cases) (Table 5).

**Table 1. Adverse event by age group, gender, dose, and time to onset (N = 95).**

| VARIABLE | | FREQUENCY | PERCENTAGE |
|---|---|---|---|
| Age Group | <20 | 3 | 3.2 |
| | 20–29 | 19 | 20.0 |
| | 30–39 | 23 | 24.2 |
| | 40–49 | 21 | 22.1 |
| | 50–59 | 19 | 20.0 |
| | ≥60 | 10 | 10.5 |
| Gender | Male | 46 | 48.4 |
| | Female | 49 | 51.6 |
| Vaccine Dose | First | 86 | 90.5 |
| | Second | 9 | 9.5 |
| Time to Onset | Less than 6hrs | 5 | 5.3 |
| | 6-24hrs | 19 | 20 |
| | Day 1 | 56 | 58.9 |
| | Day 2 | 12 | 12.6 |
| | Day 3 | 3 | 3.2 |

**Table 2. Adverse event type, severity, intervention, and outcome.**

| ADVERSE EVENT | | FREQUENCY | PERCENTAGE |
|---|---|---|---|
| Type | Local without systemic | 4 | 4.2 |
| | Systemic without local | 57 | 60.0 |
| | Both local and Systemic | 34 | 35.8 |
| | Local with/without Systemic | 38 | 40.0 |
| | Systemic with/without Local | 91 | 95.8 |
| Severity | Mild | 77 | 81.1 |
| | Moderate | 16 | 16.8 |
| | Severe | 2 | 2.1 |
| Intervention | Observation | 76 | 80.0 |
| | Analgesic/Antipyretic | 9 | 9.5 |
| | Other outpatient treatment | 8 | 8.4 |
| | Admission | 2 | 2.1 |
| Outcome | Full Resolution | 94 | 98.9 |
| | Resolution with sequelae | 1 | 1.1 |
| Time to Outcome (Days) | 1 | 25 | 26.3 |
| | 2 | 43 | 45.3 |
| | 3 | 22 | 23.1 |
| | 4–6 | 3 | 3.2 |
| | ≥7 | 2 | 2.1 |

**Table 3. Specific local and systemic adverse event.**

| AE TYPE | SPECIFIC AE | Yes n(%) | No n(%) |
|---|---|---|---|
| LOCAL (n = 38) | Pain at the Injection Site | 38 (100) | 0 (0) |
| | Itching | 1 (2.6) | 37 (97.4) |
| | Rashes/Patches | 1 (2.6) | 37 (97.4) |
| SYSTEMIC (n = 91) | Headache | 49 (51.6) | 42 (44.2) |
| | Fever | 45 (49.5) | 46 (50.5) |
| | Gen Body weakness | 37 (40.7) | 54 (59.3) |
| | Gen Body pain | 21 (23.1) | 70 (76.9) |
| | Dizziness | 3 (3.3) | 88 (96.7) |
| | Tiredness | 10 (11.0) | 81 (89) |
| | Joint pain | 8 (8.8) | 83 (91.2) |
| | Muscle pain | 1 (1.1) | 90 (98.9) |
| | Gen Body itching | 3 (3.3) | 88 (96.7) |
| | Chest pain | 2 (2.2) | 89 (97.8) |
| | Chills/Rigor | 10 (11.0) | 81 (89.0) |
| | Nausea/Vomiting | 2 (2.2) | 89 (97.8) |
| | Breathing Difficulty | 1 (1.1) | 90 (98.9) |
| | Sleep problem | 3 (3.3) | 88 (96.7) |
| | Appetite problem | 4 (4.4) | 87 (95.6) |
| | Abnormal sensation | 3 (3.3) | 88 (96.7) |
| | Difficulty in Swallowing | 2 (2.2) | 89 (97.8) |

AE = Adverse event.

**Table 4. Adverse event categories based on age, gender, dose, and time to onset.**

| Variable | AE TYPE Local Yes | No local | X²* | P-value* | Systemic Yes | No | X²* | P-value* |
|---|---|---|---|---|---|---|---|---|
| Age group | | | | | | | | |
| <20 | 2 (50) | 2 (50) | | | 4 (100) | 0 (0) | | |
| 21–40 | 16 (38.1) | 26 (61.9) | | | 41 (97.6) | 1 (2.4) | | |
| 41–60 | 16 (38.1) | 26 (61.9) | | | 40 (95.2) | 2 (4.8) | | |
| >60 | 4 (57.1) | 3 (42.9) | 1.126 | 0.771 | 6 (85.7) | 1 (14.3) | 1.896 | 0.594 |
| Gender | | | | | | | | |
| Male | 17 (37.0) | 29 (63.0) | | | 44 (95.7) | 2 (4.3) | | |
| Female | 21 (42.9) | 28 (57.1) | 0.344 | 0.557 | 47 (95.9) | 2 (4.1) | 0.004 | 1.000 |
| Vaccine dose | | | | | | | | |
| 1st | 33 (38.4) | 53 (61.6) | | | 83 (96.5) | 3 (3.5) | | |
| 2nd | 5 (55.6) | 4 (44.4) | 1.002 | 0.476 | 8 (88.9) | 1 (11.1) | 1.174 | 0.333 |
| Time to onset | | | | | | | | |
| Day 0 | 7 (29.2) | 17 (70.8) | | | 24 (100) | 0 (0) | | |
| Day 1 | 24 (42.9) | 32 (57.1) | | | 53 (94.60) | 3 (5.4) | | |
| Day 2 | 5 (41.7) | 7 (58.3) | | | 11 (91.7) | 1 (8.3) | | |
| Day 3 | 2 (66.7) | 1 (33.3) | 3.069 | 0.689 | 3 (100) | 0 (0) | 2.889 | 0.717 |

* Fisher's exact test and likelihood ratio were reported when greater than 20% of cells have expected frequencies of less than 5 for 2x2 tables and larger than 2x2 tables respectively.

AE = Adverse event.

## Discussion

Vaccines have a long history of being a cost-efficient and highly effective public health intervention for disease prevention. Vaccination is, therefore, one of the most efficient approaches

**Table 5. Adverse event severity based on age, gender, dose and time to onset.**

| Variable/Severity | Mild | Moderate | Severe | X²* | P-value* |
|---|---|---|---|---|---|
| Age group | | | | | |
| <20 | 4 (100) | 0 (0) | 0 (0) | | |
| 21–40 | 30 (73.2) | 9 (21.9) | 2 (4.9) | | |
| 41–60 | 34 (85.0) | 6 (15.0) | 0 (0) | | |
| >60 | 9 (90.0) | 1 (10.0) | 0 (0) | 5.869 | 0.438 |
| Gender | | | | | |
| Male | 36 (78.2) | 9 (19.6) | 1 (2.2) | | |
| Female | 41 (83.7) | 7 (14.3) | 1 (2.0) | 0.481 | 0.786 |
| Vaccine dose | | | | | |
| 1st | 70 (81.4) | 14 (16.3) | 2 (2.3) | | |
| 2nd | 7 (77.8) | 2 (22.2) | 0 (0) | 0.568 | 0.753 |
| Time to onset | | | | | |
| Day 0 | 16 (66.7) | 8 (33.3) | 0 (0) | | |
| Day 1 | 49 (87.5) | 5 (8.9) | 2 (3.6) | | |
| Day 2 | 10 (83.3) | 2 (16.7) | 0 (0) | | |
| Day 3 | 2 (66.7) | 1 (33.3) | 0 (0) | 10.082 | 0.433 |

* Likelihood ratio used when greater than 20% of cells have expected frequencies of less than 5.

to illness prevention [22]. Vaccination hesitancy, on the other hand, can seriously harm it [6]. The degree of uptake in the population has a big impact on the effectiveness of immunisation in preventing COVID-19. Herd immunity is achieved when a large number of people get vaccinated [16]. People are concerned about the COVID-19 vaccine's potential for adverse events and hazards [23]. Due to their lower frequency, the smaller number of persons participating in studies, and other comparable constraints, some adverse events may not have been observed in pre-clinical trials. As a result, post-vaccination adverse event monitoring is crucial for alerting the public and policymakers about the vaccine's safety and the potential for severe reactions. This study examined the adverse events people had after getting the AstraZeneca COVID-19 vaccine in five vaccination centers in Kwara State, Nigeria.

The immunization was followed by adverse events in 95 (1.6%) of the 5816 people who received it. Approximately a quarter of the adverse events (25.3 percent) developed within 24 hours of immunization. Although a vast percentage of vaccine recipients (98.4%) reported no adverse event, this does not imply that the vaccine was unsuccessful. Despite the fact that another type of COVID vaccine (Moderna) protected 95% of people who received it from the virus, just 10% of those who received it reported adverse events [24]. The immune system of every individual reacts differently to vaccines, with some having more physical response to the vaccine than others. Some reasons for this difference are gender, age, individual's health status, time of the day that vaccine was received, individual's nutrition, and environment [12].

The incidence of adverse events in this study is far lower than 79.4% and 90.9% reported for the same vaccine in Nepal and Korea respectively [20,25]. Other studies found 93.5% incidence in Afghanistan and 90.1% in South-south Nigeria [12,23]. A possible explanation for the wide disparity in the incidence of adverse events in these studies compared to ours is that all of these studies were carried out among health care workers whereas, ours was carried out among the general population. Poor reporting culture of adverse events may be another reason for the disparity. Although low reporting of adverse events is a global phenomenon, the Nigerian situation is worse [26]. Despite the crucial role of patients in spontaneous reporting of adverse events [27], there exists a general belief in Nigeria's traditional communities that adverse events are proof of the effectiveness of medicinal agents [28].

In our investigation, the majority of adverse events were mild to moderate in severity.

There were no deaths among those who received the vaccination. Although two cases of severe adverse events were reported, only one was serious and occurred in a 30-year-old woman who received the first dosage. She developed anaphylactic shock, hyperpyrexia, severe headache, loss of taste, chest discomfort, difficulty breathing, pain at the injection site, and was unable to communicate. She was resuscitated and initially treated in the emergency room, then received subsequent treatment and was discharged after 48 hours of full recovery. There were no sequelae after 3 months of follow-up investigation.

AE severity in our study was mild to moderate, which is consistent with practically all prior investigations of AstraZeneca COVID-19 vaccine [12,20,23,25,29]. For instance, none of the AstraZeneca COVID-19 vaccine recipients in Nepal developed severe adverse events or required long-term hospitalization and no deaths were reported within a week [20]. Similarly, only one incidence of acute allergic reaction with hypotension was documented in the South Korean research, which subsided spontaneously and there was no significant AE that necessitated hospitalization or resulted in death [29].

These mild-to-moderate AE are allowable during COVID-19 vaccination. These findings serve to alleviate vaccine apprehension generated by fears of serious adverse events from the COVID-19 vaccine. The majority of the adverse events (80%) subside spontaneously within two days (71.6%). These observations are considered typical and indicate that the body is establishing an immunological reaction against COVID-19. There was no evidence of blood

clots, pulmonary embolism, deep vein thrombosis, focal neurological deficit, or cardiac event in this study's participants within 3 months of follow-up. However, the duration of follow-up may not permit definitive comment on the long-term adverse event of the vaccine.

The majority of the reported adverse events (95.8%) had a combination of the two reactions which were either systemic and/or local. Injection-site pain occurred in all the individuals with local reactions while itching and rashes at the injection site were the least occurring ones. The most commonly reported systemic reactions were headache (51.6%), fever (49.5%), generalized body weakness (40.7%), and generalized body pain (23.1%). This finding is consistent with the result of analysis of combined data of AstraZeneca clinical trials conducted in South Africa, Brazil, and the United Kingdom where the frequently occurring adverse events were injection site pain, fatigue, headache, malaise, myalgia, and fever. The majority of adverse events in those trials were also found to be mild to moderate in severity and resolved soon after the vaccination [14]. Azimi et al [23] from Afghanistan also reported injection site pain as the most common local adverse event to the AstraZeneca vaccine while muscle pain, fever as well as fatigue were the most common systemic adverse event. He also reported close to half of the vaccine recipients have a headache, joint pain, and chills.

Furthermore, the commonest adverse event from the AstraZeneca vaccine in Yenagoa South-south Nigeria was pain at the injection site, followed by fever (57%) while other adverse events were headache, fatigue, chills, muscle pains, and joint pains [12]. These have also been reported in other previous studies [14,20,23,29]. There was no significant association between the age, gender, vaccine dose as well as time to onset of AE, and the reported adverse event categories in terms of local and systemic reactions. Also, no significant association was found between the adverse event severity and age, gender, vaccine dose as well as time to onset of adverse event. Similarly, analysis of socio-demographic and clinical parameters did not reveal any statistically significant predictor for all the AEs tested among AstraZeneca vaccine recipients in Ilorin North central Nigeria [19]. This may suggest that most AE are often unpredictable. More studies will be required to further demonstrate the associated factors of COVID-19 AE severity and categories in the general population.

## Limitation

The reliability of self-reported symptoms may be highly subjective and not depict the real adverse event experienced. The poor reporting culture of adverse events among Nigerians may also conceal the actual figure of the adverse events.

## Conclusions

In the face of an ever-changing circumstance, it seems self-evident that vaccination is still the greatest way to achieve herd immunity. This is especially critical in Nigeria, where only 2.6 percent of the population was fully vaccinated as of February 1, 2022 [4]. This is a shockingly low figure. Efforts to assuage anxieties and concerns regarding vaccine adverse events should continue to be made through health education and promotion. This study was significant in determining the adverse events from the AstraZeneca vaccination in the general population. Concerns about COVID-19 vaccine safety and adverse events can impact vaccine uptake, therefore this finding is especially important. Furthermore, the findings of this study have offered crucial data for a better understanding of COVID-19 vaccination-related AE and will serve as a foundation for future research.

## Supporting information

**S1 Dataset.**
(SAV)

## Acknowledgments

We will like to appreciate all the staff working in the five immunization centers for their support during the conduct of this study.

## Author Contributions

**Conceptualization:** Louis Okeibunor Odeigah, Yahkub Babatunde Mutalub, Olalekan Ayodele Agede, Ismail A. Obalowu, Susan Aiyetoro, Gafar A. A. Jimoh.

**Data curation:** Yahkub Babatunde Mutalub, Susan Aiyetoro.

**Formal analysis:** Yahkub Babatunde Mutalub, Susan Aiyetoro.

**Project administration:** Louis Okeibunor Odeigah, Ismail A. Obalowu, Gafar A. A. Jimoh.

**Supervision:** Louis Okeibunor Odeigah, Ismail A. Obalowu, Gafar A. A. Jimoh.

**Visualization:** Yahkub Babatunde Mutalub.

**Writing – original draft:** Yahkub Babatunde Mutalub.

**Writing – review & editing:** Louis Okeibunor Odeigah, Yahkub Babatunde Mutalub, Olalekan Ayodele Agede, Ismail A. Obalowu, Gafar A. A. Jimoh.

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
