## [Decision Letter · Decision Letter 0]

3 May 2022

PGPH-D-22-00480

Adverse events following COVID-19 vaccination in Kwara state, North-central Nigeria.

Dear Dr. Mutalub,

Thank you for submitting your manuscript to PLOS Global Public Health. After careful consideration, we feel that it has merit but does not fully meet PLOS Global Public Health’s publication criteria as it currently stands. Therefore, we invite you to submit a revised version of the manuscript that addresses the points raised during the review process.

Please submit your revised manuscript by . If you will need more time than this to complete your revisions, please reply to this message or contact the journal office at globalpubhealth@plos.org. Please include the following items when submitting your revised manuscript:

We look forward to receiving your revised manuscript.

Kind regards,

Abram Luther Wagner, PhD, MPH

Academic Editor

Journal Requirements:

1. We have amended your Competing Interest statement to comply with journal style. We kindly ask that you double check the statement and let us know if anything is incorrect. 

2. We ask that a manuscript source file is provided at Revision. Please upload your manuscript file as a .doc, .docx, .rtf

4. Please provide an Author Summary. This should appear in your manuscript between the Abstract (if applicable) and the Introduction, and should be 150–200 words long. The aim should be to m+E35ake your findings accessible to a wide audience that includes both scientists and non-scientists. Sample summaries can be found on our website under Submission Guidelines:

https://journals.plos.org/globalpublichealth/s/submission-guidelines#loc-parts-of-a-submission

Additional Editor Comments (if provided):

Please see the comments below and respond as thoroughly as you can. In your response to reviewers, please explicitly mention what you have changed and where it could be found in your manuscript.

Reviewers' comments:

Reviewer's Responses to Questions

**Comments to the Author**

1. Does this manuscript meet PLOS Global Public Health’s publication criteria? Is the manuscript technically sound, and do the data support the conclusions? The manuscript must describe methodologically and ethically rigorous research with conclusions that are appropriately drawn based on the data presented.

Reviewer #1: Yes

Reviewer #2: Partly

2. Has the statistical analysis been performed appropriately and rigorously?

Reviewer #1: Yes

Reviewer #2: Yes

3. Have the authors made all data underlying the findings in their manuscript fully available (please refer to the Data Availability Statement at the start of the manuscript PDF file)?

Reviewer #1: Yes

Reviewer #2: Yes

4. Is the manuscript presented in an intelligible fashion and written in standard English?

Reviewer #1: Yes

Reviewer #2: Yes

5. Review Comments to the Author

Reviewer #1: This is a well written paper, and the work will contribute to the important area of vaccine safety reporting and the role it plays in mitigating vaccine hesitancy.

1. In the methods, there is no mention of which vaccine(s) is/are being monitored but paragraph 2 of the discussion talks about AstraZeneca and Morderna. Authors need to clarify on which vaccine to which the AEFIs are attributed

2. Was this a document review of records at the health facilities? Or the vaccine recipients were interviewed using a dedicated tool by health worker or research assistant?

3. Table 4 and 5: We need to be careful when interpreting the chi-square and p-values because there are many cells with zero or n<5. Did authors use Fisher’s exact test?

4. Need to be consistent with the use of ‘adverse events’, ‘adverse reactions’, ‘negative reactions’ and ‘side effects.’ They are used interchangeably, and authors need to harmonise that

Discussion

4. Paragraph 3 of discussion is confusing. The reasons given for fewer numbers are .......general population and being community-based health care facilities..... These reasons may not be convincing to the general audience. I think the AE reporting culture may be one of the reasons …. Please explore further to make your argument stronger

5. Para 4: what do you mean by this statement ‘……If age is a factor of AE, there is no clear consensus.’ A lot of contradicting information from sentence number 4 onwards.

6. Para 5: gender comparison would be stronger if the number vaccinated is known by gender.

7. Para 6: the last sentence seems to be incomplete

8. Para 7: the last 2 sentences may be seen as subjective because the follow-up time is not mentioned anywhere yet….

Reviewer #2: The author described the fear of adverse effects of vaccination in general public resulting hesitancy and resistance of vaccination. Moreover the lower number are been vaccinated in the particular state. The evaluation was done by taking only one vaccine i.e. AstraZeneca, what about other vaccines side effects/AE?. The interventions made in case of fatal AE that involved hospitalization were not clearly described the outcomes. What kind of observations made in interventions?. The limitations of study showed that ADE might not be depicted appropriately.

The article is well written, describing some points that should be helpful for considerations in future studies. However, it is important to compare the data on severity of adverse effects of one or more vaccines that have been used in the state.

6. PLOS authors have the option to publish the peer review history of their article (what does this mean?). If published, this will include your full peer review and any attached files.

**Do you want your identity to be public for this peer review?** For information about this choice, including consent withdrawal, please see our Privacy Policy.

Reviewer #1: No

Reviewer #2: No

---

## [Decision Letter · Decision Letter 1]

23 Jun 2022

PGPH-D-22-00480R1

Adverse events following COVID-19 vaccination in Kwara state, North-central Nigeria.

Dear Dr. Mutalub,

Thank you for submitting your manuscript to PLOS Global Public Health. After careful consideration, we feel that it has merit but does not fully meet PLOS Global Public Health’s publication criteria as it currently stands. Therefore, we invite you to submit a revised version of the manuscript that addresses the points raised during the review process.

Please submit your revised manuscript by . If you will need more time than this to complete your revisions, please reply to this message or contact the journal office at globalpubhealth@plos.org. Please include the following items when submitting your revised manuscript:

We look forward to receiving your revised manuscript.

Kind regards,

Abram L. Wagner, PhD, MPH

Academic Editor

Journal Requirements:

Additional Editor Comments (if provided):

The reviewer has indicated that you have adequately responded to their comments. Could you make the following changes? After you re-submit, I could quickly process this manuscript.

1. Could you separate the first sentence of the abstract into two sentences?

2. For the discussion first two sentences - could you reword? I would hesitate to say that vaccination is the most efficient approach to illness prevention. (you could say "vaccination is one of the most efficient approaches to illness prevention").

3. You focus in the discussion on the difference in AE incidence between males and females. However, I don't think you found a significantly different result. If you really want to discuss this, I would start the paragraph saying something like "We found a non-significantly higher proportion of AEs in woman than men. [other studies...]"

4. I would also hesitate to make comparisons by demographic group. Your Table 1 is good, but to adequately compare the risk of AE, we would need to know what these numbers are in the total 5816 individuals who were vaccinated. That is to say, I would recommend not focusing much on age or gender - you could keep those paragraphs in discussion or delete them. Your discussion about the types of AEs was more interesting and relevant given your data.

5. You note a very low proportion of AEs in your study. Your description seems adequate, but I would move it to the limitations section.

Reviewers' comments:

Reviewer's Responses to Questions

**Comments to the Author**

1. If the authors have adequately addressed your comments raised in a previous round of review and you feel that this manuscript is now acceptable for publication, you may indicate that here to bypass the “Comments to the Author” section, enter your conflict of interest statement in the “Confidential to Editor” section, and submit your "Accept" recommendation.

Reviewer #1: All comments have been addressed

2. Does this manuscript meet PLOS Global Public Health’s publication criteria? Is the manuscript technically sound, and do the data support the conclusions? The manuscript must describe methodologically and ethically rigorous research with conclusions that are appropriately drawn based on the data presented.

Reviewer #1: Yes

3. Has the statistical analysis been performed appropriately and rigorously?

Reviewer #1: Yes

4. Have the authors made all data underlying the findings in their manuscript fully available (please refer to the Data Availability Statement at the start of the manuscript PDF file)?

Reviewer #1: Yes

5. Is the manuscript presented in an intelligible fashion and written in standard English?

Reviewer #1: Yes

6. Review Comments to the Author

Reviewer #1: (No Response)

7. PLOS authors have the option to publish the peer review history of their article (what does this mean?). If published, this will include your full peer review and any attached files.

**Do you want your identity to be public for this peer review?** For information about this choice, including consent withdrawal, please see our Privacy Policy.

Reviewer #1: No

---

## [Editor Report · Decision Letter 2]

6 Jul 2022

Adverse events following COVID-19 vaccination in Kwara state, North-central Nigeria.

PGPH-D-22-00480R2

Dear Dr Mutalub,

We are pleased to inform you that your manuscript 'Adverse events following COVID-19 vaccination in Kwara state, North-central Nigeria.' has been provisionally accepted for publication in PLOS Global Public Health.

Best regards,

Abram L. Wagner, PhD, MPH

Academic Editor